

# Age and driving mechanisms of the Eocene-Oligocene Transition from astronomical tuning of a lacustrine record (Rennes Basin, France)

Slah Boulila[1,2], Guillaume Dupont-Nivet[3,4], Bruno Galbrun[1], Hugues Bauer[5], Jean-Jacques Châteauneuf[5]

[1]Sorbonne Université, CNRS , Institut des Sciences de la Terre-Paris, ISTeP, F-75005 Paris, France
[2]ASD/IMCCE, CNRS-UMR 8028, Observatoire de Paris, PSL University, Sorbonne Université, Paris, France
[3]Geosciences Rennes UMR 6118, Université Rennes 1, CNRS, Rennes, France
[4]Potsdam University, Department of Earth and Environmental Sciences, Potsdam-Golm, Germany
[5]BRGM, Bureau de Recherches Géologiques et Minières, Orléans, France

*Correspondence to*: Slah Boulila (slah.boulila@sorbonne-universite.fr)

**Abstract.** The Eocene-Oligocene Transition (EOT) marks the onset of the Antarctic glaciation and the switch from greenhouse to icehouse climates. However, the driving mechanisms and the precise timing of the EOT remain controversial mostly due to the lack of well-dated stratigraphic records, especially in continental environments. Here we present a cyclo-magnetostratigraphic and sedimentological study of a ~7.6 Myr-long lacustrine record spanning the late Eocene to the earliest Oligocene, from a drill-core in the Rennes Basin (France). Time-series analysis of natural gamma-ray (NGR) log data shows evidence of Milankovitch cycle bands. In particular, the 405 kyr stable eccentricity is expressed with strong amplitudes. Astronomical calibration to this 405 kyr periodicity yields duration estimates of Chrons C12r through C16n.1n, providing additional constraints on the middle–early Eocene timescale. Correlations between the orbital eccentricity curve and the 405 kyr tuned NGR time series and assumptions on their phase relationships, enable to test previously proposed ages for the EO boundary, indicating that 33.71 and 34.10 Ma are the most likely. Additionally, the 405 kyr tuning calibrates the most pronounced NGR cyclicity to a period of ~1 Myr matching the g1-g5 eccentricity term. Such cyclicity has been recorded in other continental records, pointing to its significant expression in continental depositional environments. The record of g1-g5 and sometimes g2-g5 eccentricity terms in previously acquired sedimentary facies proxies in CDB1 core led us to hypothesize that the paleolake level may have behaved as a lowpass filter for orbital forcing. Two prominent changes in the sedimentary facies were detected across the EOT, which are temporally equivalent to the two main climatic steps, EOT-1 and Oi-1. Combined with previously acquired geochemical ($\delta^{15}N_{org}$, TOC), mineralogical (Quartz, clays) and pollen assemblage proxies from CDB1, we suggest that these two facies changes reflect the two major Antarctic cooling/glacial phases via the hydrological cycle, as significant shifts to drier and cooler climate conditions, thus supporting the stepwise nature of the EOT. Remarkably, a strongly dominant obliquity expressed in the latest Eocene corresponds in time to the interval from the EOT precursor glacial event till the EOT-1. We interpret the obliquity dominance as reflecting preconditioning phases for the onset of the major Antarctic glaciation, either from its direct impact on the





formation/(in)stability of the incipient Antarctic Ice Sheet (AIS), or through its modulation of the North Atlantic Deep Water production given the North Atlantic coastal location of the CDB1 site.

## 35  1 Introduction

The Eocene-Oligocene climate transition (EOT) is one of the most drastic climate changes of the Cenozoic era and the final stage of the switch from greenhouse to icehouce climates. It is characterized by the onset of large and perennial ice sheets on Antarctica (e.g., Miller et al., 1991; Zachos et al., 2001a) inducing global cooling and significant sea-level lowering (Lear et al., 2008; Miller et al., 2008; Hren et al., 2013; Goldner et al., 2014), a deepening of the calcite compensation depth (Coxall
et al., 2005; Kaminski and Ortiz, 2014), and severe and widespread disturbances in biotic, carbon, and hydrological cycles (e.g., Dupont-Nivet et al., 2007; Pearson et al., 2009; Coxall and Wilson, 2011; Xiao et al., 2010; Coxall et al., 2018; Hutchinson et al., 2021). Data from deep-sea carbon and oxygen stable isotopes particularly suggest that the EOT occurred in successive steps (Coxall et al., 2005; Katz et al., 2008; Coxall and Wilson, 2011), and that the Antarctic glaciation was initiated, after climatic preconditioning by a decline in atmospheric $CO_2$ enhanced by a specific orbital configuration of
synchronous eccentricity and obliquity minima (Zachos et al., 2001b; DeConto and Pollard, 2003; Coxall et al., 2005; Pearson et al., 2009).

Precise timing of the EOT is therefore crucial to understand the significance of the successive and short-lived climatic events occuring during this time interval. Astronomical tuning of the geologic time is constantly progressing, and is today towards its achievement for the Cenozoic era. However, the recent Geologic Time Scales GTS2012, GTS2016 and
GTS2020 (Vandenberghe et al., 2012; Ogg et al., 2016; Speijer et al., 2020) are still lacking precise astronomical tuning in the middle-late Eocene interval, described therein as the "*Eocene astronomical time scale gap*". Although important efforts to fill this gap differences between published age models still persist (Speijer et al., 2020).  In particular, the age of the Eocene–Oligocene boundary (EOB), which is a fundamental tie-point for the Cenozoic timescale, remains controversial (see Hilgen and Kuiper, 2009 for a review, and a later study by Sahy et al., 2017).

Sedimentary records and climate modeling have demonstrated that lacustrine environments are sensitive areas to solar radiation change induced by Milankovitch orbital forcing (e.g., van Vugt et al., 1998; Abels et al., 2010). Although lacustrine records are known to be more affected by sedimentary discontinuities than deep-sea sequences, numerous cyclostratigraphic studies have shown their usefulness for astronomical calibration of the geologic timescale (e.g., van Vugt et al., 1998; Xiao et al., 2010; Hong et al., 2020).

Important investigations have been focused on marine environments to document the EOT, in particular from deep-sea drilling programs (e.g., Coxall and Wilson, 2011 and references therein), but fewer are the studies from continental records (e.g., Dupont-Nivet et al., 2007; Xiao et al., 2010; Sun et al., 2014; Toumoulin et al., 2021). High-resolution records of the EOT in continental environments are needed to constrain the atmospheric expression of the EOT, which is essential for climate modeling and to extend our understanding of its forcing mechanisms.



Here we investigate integrated cyclo-magnetostratigraphy and sedimentology on a late Eocene to earliest Oligocene lacustrine record from a core drilled in 2010 in the Rennes Basin in Brittany, northwestern France (Bauer et al., 2016; Fig. 1). The record covers with high fidelity the stratigraphic interval spanning Chrons C12r through C16n.1n. The sedimentary record exceptionally captures the EOT in a continuous and steady lacustrine-palustrine environment. Three main objectives of the present study are: (1) to investigate whether the cyclic lacustrine deposits are astronomically-driven, (2) then to

provide astronomical calibration of the recovered polarity chrons for fine tuning the Paleogene timescale, and (3) to compare this rare mid-latitude continental record with reference marine records in order to better understand the forcing mechanisms of the EOT.

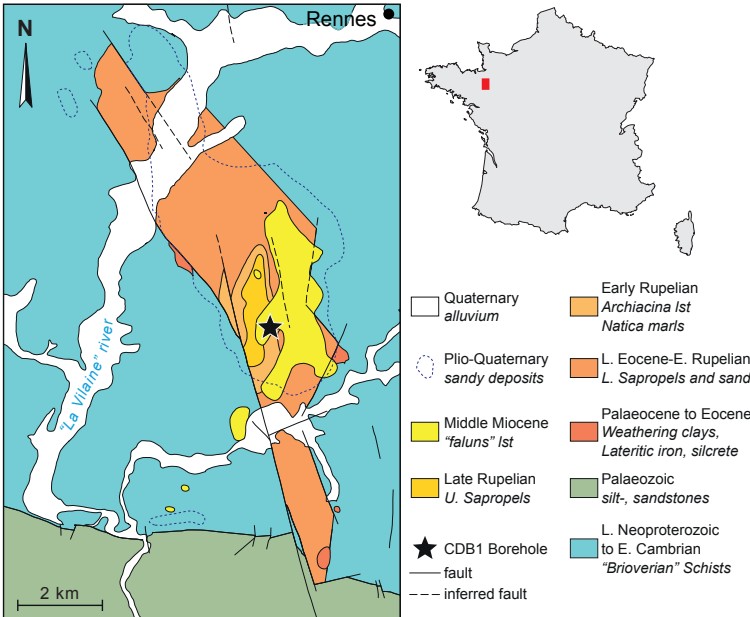


**Figure 1: Geographic and geologic location of CDB1 drill-core in the Rennes Basin (NW France). Position of CDB1 core is shown by a black star (modified from Bauer et al., 2016).**




## 2 Geologic setting and stratigraphic framework

### 2.1 Geological setting of the Rennes Basin

The Rennes Basin is the deepest of numerous small Cenozoic basins scattered across the Armorican Massif (Northwestern France). It acted as a transtensive basin during the Eocene and early Oligocene before re-acting as a transpressive basin since

the Miocene (Bauer et al., 2016 and references therein). The 675 m long CDB1 borehole crossed all of the Rennes basin's formations, down to the deeply weathered basement (Bauer et al., 2016). Description of all formations was provided by Bauer et al. (2016). The Rennes Basin is filled up with more than 400 m of Cenozoic deposits, identified as six lithologic formations (Tables S1 and S2). It was initiated as a fluvio-lacustrine depositional environment, with rare marine incursions attesting to a coastal setting of the Chartres-de-Bretagne Formation (Bauer et al., 2016), then alternating lacustrine and

palustrine depositional settings attributed to the Lower Sapropels, deposited in dysoxic/anoxic environments that preserved organic matter accumulations (Bauer et al., 2016; Tramoy et al., 2016). The studied sedimentary interval includes the Natica Marls (66-85.15m), the Lower Sapropels (85.15-375.40m) and the Chartres-de-Bretagne (375.40-404.92m) Formations (Fig. 2 and Table S1).

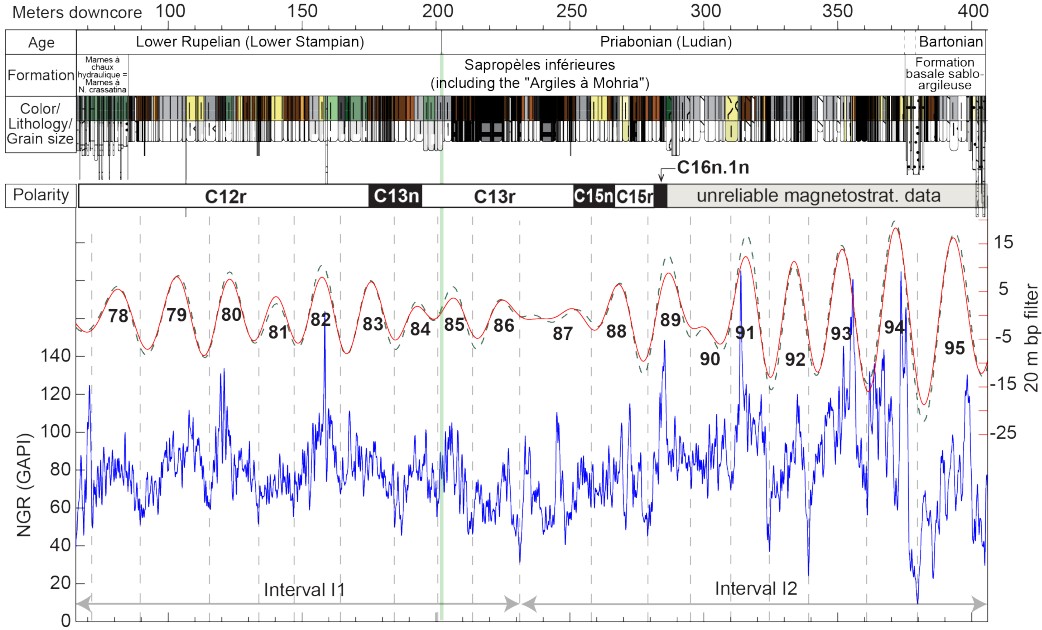


**Figure 2: Integrated cyclo-magnetostratigraphy of the CDB1 drill-core. Paleomagnetic data are provided in the Supplementary Information. For sediment color, lithology and grain size, see Bauer et al., 2016. On the natural gamma-ray (NGR) data (10 cm**





**resolution) a gaussian bandpass filter (0.05 ±0.015 cycles/m in solid line, 0.05 ±0.02 cycles/m in dashed line) is applied to extract the**
**prominent cyclicity of a ~20 m wavelength, interpreted as reflecting the 405 kyr eccentricity periodicity. The numbering scheme**
**for the 405 kyr eccentricity cycles follows the approach of Hinnov and Hilgen (2012), where eccentricity maxima are counted back**
**in time. Intervals I1 and I2 correspond to those used for time-series analysis in depth domain (Fig. 3).**

## 2.2 Biostratigraphic framewok of the CDB1 core

Biostratigraphic constraints are provided by nearly one hundred stratigraphic levels sampled and analysed throughout the core (Table S1; Bauer et al., 2016). The biostratigraphy relies on various fossils and microfossils, including malacofaunal assemblage, benthic foraminifera, dinocyst and pollen grain assemblages (Table S2). But most of the studied intervals rely on pollen constraints that are refined by comparison with well-dated basins from Western Europe, especialy with the nearby Paris Basin (see Bauer et al., 2016 and references therein). Depending on recovered fossil assemblages and sampling

resolution, uncertainties on estimated age or biozone boundaries vary, ranging from less than 1 m for the top SBZ21 biozone to ca. 110 m for the Bartonian/Priabonian boundary (Table S2). The Priabonian/Rupelian boundary (Eocene-Oligocene boundary) is relatively well constrained by a marked biozone boundary bracketed between 195.08 and 205.99 m depth (Table S2).

## 3 Data and methods

### 3.1 Magnetostratigraphy

Sampling of ca. 220 standard paleomagnetic oriented cylinders (2.5-cm diameter and 4- to 10-cm long) was performed using an electric drill at regular 1-3-meter intervals throughout the whole preserved portion of the core from 66 to 411 m depth.

Remanent magnetizations of samples were measured on a 2G Enterprises DC SQUID cryogenic magnetometer within an amagnetic chamber, at the Geosciences Rennes paleomagnetic laboratory, France. A first, selection of pilot

samples distributed around 10 meters intervals throughout the core was stepwise demagnetized both thermally and using alternative field (AF). A high-resolution set of demagnetization steps were initially applied in order to (1) determine the characteristic demagnetization behaviour, (2) establish the most efficient demagnetization temperature and AF steps, (3) determine which lithology provides the best signal, and (4) identify stratigraphic intervals with potential paleomagnetic reversals. These preliminary results guided further processing of selected remaining samples at higher stratigraphic

resolution. To better identify magnetic mineralogies, Isothermal Remanent Magnetization (IRM) acquisitions separated in 3 coercivity components (1150, 400 and 125 mT) followed by thermal demagnetizations were undertaken on a set of 10 representative samples (Lowrie, 1990; Fig. S2).

Characteristic Remanent Magnetization (ChRM) directions were calculated using a minimum of four consecutive heating steps on thermal demagnetization paths displayed on vector-end point diagrams (Fig. S1). A careful selection of

ChRM directions (especially of normal directions) was performed by grouping them by quality (Fig. S1). Great-circle





analysis was applied to a few samples, when a secondary normal polarity component overlapped a reversed polarity direction carried by only a few points (the mean of the Q1 reverse polarity directions was used as set point following McFadden and Reid (1982).

### 3.2 Natural Gamma-ray (NGR)

Natural gamma-ray (NGR) in sedimentary rocks is due to gamma rays emitted via natural radioactivity from mainly potassium ($K^{40}$), thorium ($Th^{232/230}$), and uranium ($U^{238/235}$). These elements are mainly present in clay minerals and the NGR therefore essentially expresses the clay/carbonate ratio. NGR has been used successfully as a climatic proxy to detect orbitally forced sediments, and has served as a powerful tool for cyclostratigraphy and orbital tuning (e.g., van Vugt et al., 1998).

The NGR probe used during the CDB1 well logging consists of a scintillation detector (sodium iodide crystal coupled to a photo-multiplication tube), which converts gamma rays in electric pulses. The measured values are expressed in API (American Petroleum Institute) units (Fig. 2). These units are calculated using a linear interpolation of raw data expressed in pulses per second (pps) according to a calibration set with a very stable source, the radioactivity of which is known in API units. Because disintegration is a random phenomenon, raw measurments taken every 2 cm are integrated over

a 25-cm long window and restituted at a 10-cm step, which is consistent both with the crystal size and the average dip of the strata (~15°). The NGR dataset is available as supplementary data.

### 3.3 Cyclostratigraphic analysis

To quantify significant wavelengths in sedimentary proxies that could be related to specific orbital frequencies, we used the multitaper method (MTM, Thomson, 1982) associated with the robust red noise modeling (Mann and Lees, 1996). Because

of the high variability in the NGR signal throughout the core, we subdivided it into two intervals I1 and I2 (Fig. 2). Then, to better visualize both lower and higher portions of the spectra, analyses were conducted with three $2\pi$ data tapers, using a logarithmic scale for the power axis (e.g., Fig. 3). To study the evolution of cycles throughout the record, we used evolutive harmonic analysis (EHA) with a function computing a running periodogram of a uniformly sampled (dx=10 cm) time series using FFTs of zero-padded segments, and normalized to the highest amplitudes (e.g., Kodama and Hinnov, 2014; Fig. S8).

To extract target cycles, we used conjointly bandpass gaussian filtering (Paillard et al., 1996) and the smoothing technique based on the weighted average LOWESS method (e.g., Fig. 3). Finally, to convert the depth into the time domain, we used the 405 kyr stable eccentricity period because it is well expressed along the whole core (Figs. 2 and S8), and its phase is very stable (Laskar et al., 2004, 2011). We anchored the NGR record at previously proposed ages of EOB. Then, we established several potential correlations between the 405 kyr cycles from NGR and La2011 astronomical model, in order to seek for the

best fit using the cross-MTM spectral analysis (e.g., Huybers and Denton, 2008). We have aso tested the continuity of CDB1 Core by comparing the 405 kyr tuning with 21 kyr tuning (precession mean period, Fig. S7C).





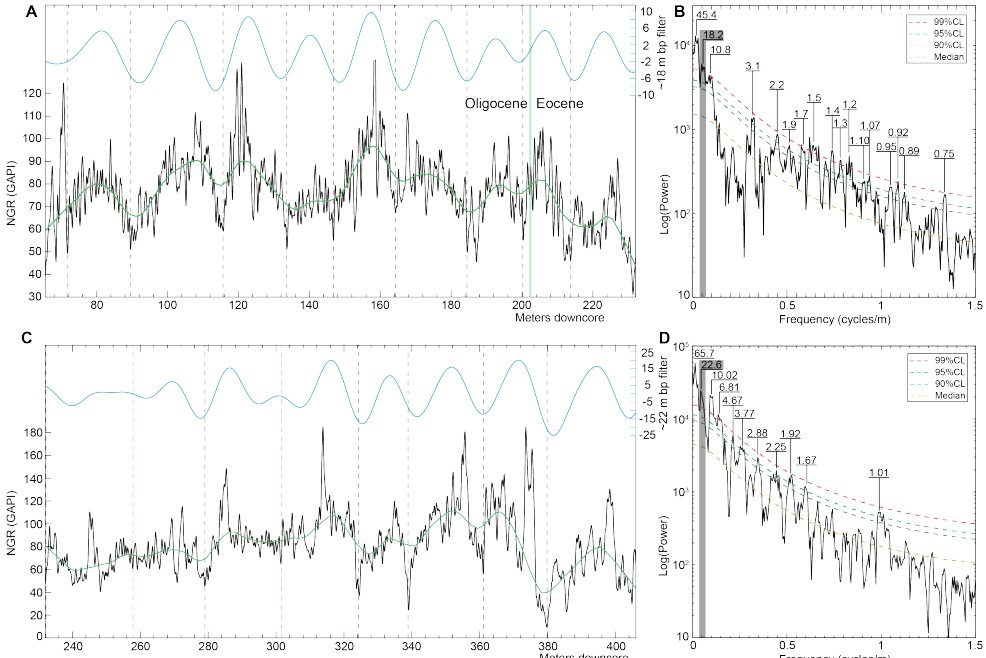

**Figure 3: Time-series analysis per intervals I1 (65.5 to 232 m) and I2 (232 to 406 m) (see Fig. 2) of the untuned NGR variations. (A) Interval I1, along with 18.2 m bandpass filtered (as in Fig. 2) and smoothed with a 8% weighted average of the series. (B) 2π-MTM power spectrum of interval I1. (C) Interval I2, along with 22.6 m bandpass filtered (as in Fig. 2) and smoothed with a 8% weighted average of the series. (D) 2π-MTM power spectrum of interval I2. Grey-shaded vertical bars in 'B' and 'D' correspond to the filtered cyclicity in 'A' and 'C'. All spectral periods in 'B' and 'D' are expressed in meters.**

## 4 Results

### 4.1 Rock magnetism analyses and magnetostratigraphy

Initial Natural Remanent Magnetizations (NRM) were typically low (Table S3) ranging between $10^{-3}$ to $10^{-6}$ A/m and thus requiring the use of a highly sensitive cryogenic magnetometer. Lower NRM values were generally recorded in the lower part of the core in particular below ca. 300 m depth where most samples yielded uninterpretable erratic demagnetization path that can be attributed to their low NRM values, in the order of instrumental detection. Above ca. 300 m depth, however, most of the 114 samples analyzed yielded higher NRM values usually far above the detection level of the magnetometer (Table S3) with interpretable demagnetization paths. These thermal demagnetizations diagrams (Fig. S1) generally display two main components: a low temperature component (LTC) from 0 to 150-180°C and a higher temperature component (HTC)



from 180-250 to 600°C. However, above 300-350°C, demagnetization paths often become erratic showing unstable remanences with often increasing remanence intensities and increasing magnetic susceptibilities suggesting mineral transformation in most samples. AF demagnetization yielded less separation of LTC and HTC with demagnetization becoming often erratic and noisy at high applied fields and some evidence for gyroremanence acquisition. As a result, thermal rather than AF demagnetizations were applied to the remainder of the sample set with a carefully defined set of

targeted small thermal demagnetization steps (Fig. S1). This procedure generally yielded demagnetization diagrams with clearly separated LTC and HTC although they sometimes overlapped along great circle paths on stereographic projections. LTC have exclusively normal polarity orientations suggesting a recent overprint while HTC display normal and reversed polarity orientations.

The nature of the magnetization components is further indicated by the IRM acquisition showing mostly low

coercivities (<200 mT; Fig. S2a). Its separation into 3 coercivities indicate most (70-90%) of the IRM magnetization in the low coercivities (below 125 mT), 5-20% in the intermediate (125-400 mT) and <10% in the high coercivity (400-1150 mT). The following themal demagnetization shows the following behaviors. (a) The low coercivity components are demagnetized monotonically down to 600°C with some samples mostly demagnetized at 350-400°C (Fig. S2b). (b) The intermediate coercivity have noisy demagnetizations mostly achieved at 350-400°C (Fig. S2c). (c) The high coercivity is more stable but

also demagnetized mostly at 350-400°C (Fig. S2d).

The observed behaviors lead us to interpret the remanent magnetizations are carried by a combination of iron oxides and iron sulphides. The presence of iron sulfides is strongly suggested by the organic-rich nature of the sediment and in particular, the observation of pyrite formation upon splitting of the cores and their surficial oxidation in the warehouse. Furthermore, the NRM thermal demagnetization increasing and becoming unstable in the 250-400°C range are characteristic

of iron sulphide oxidation thermally transforming into iron oxides upon thermal treatment (e.g. Dunlop and Özdemir, 1997). Gyroremanence acquisition during AF treatment of NRM potentially suggest greigite (Vasiliev et al., 2008; Sagnotti et al., 2010). However, the demagnetization of samples up to 600°C indicates iron oxides also contribute to the remanence. This is further confirmed by IRM experiments. The low coercivity component carrying most of the signal is demagnetized up to 600°C suggesting magnetite. This low coercivity component may also parlty includes iron sulphides such as thermally stable

single domain greigite and pyrrhotite have coercivities below 200 mT that may be secondary or primary (Dekkers et al., 1988; Roberts et al., 1998). The intermediate and higher coercivities components demagnetized below 400°C may correspond to partial oxidation of phyrrotite and greigite acquired during diagenetic weathering or more likely during the thermal demagnetization breaking down iron sulphides, which could explain the unstability observed in this range (Roberts et al., 2011).

These consideration help interpret the observed LTC and HTC components. The LTC exclusively in a normal polarity direction can be simply explained by secondary iron sulphides formed during diagenetic processes (e.g. Roberts et al., 2011). The HTC is more complex as it likely relates to a combination of iron sulphides that may be primary or secondary and oxides that are likely primary. The magnetite present in the samples must contribute a large part of the NRM in the HTC





range but it is blurred by iron sulphides that oxidize at these temperatures becoming unstable and acquiring secondary

magnetizations that explain the difficulty to isolate the primary components in the HTC. Nevertheless, a stable HTC was often isolated after the LTC and before the directions became unstable at high temperatures (Fig S1). This HTC is most likely carried by the magnetite present in the samples as indicated by stable thermal demagnetizations up to 600°C in a few samples with little iron sulphides. However, some of the HTC may be carried by iron sulphides making the demagnetization at times difficult to interpret such that careful inspection and defining quality criteria were applied to select HTC for further

ChRM direction and polarity determinations (see Fig. S1 for the definition of the quality criteria).

The carefully selected ChRM directions obtained from the HTC yielded inclinations displaying two distinct clusters around +50° and -50° respectively as expected at this latitude and thus further suggest a primary record of normal and reversed polarities (Fig. S3). In order to remove widely outlying and transitional directions, the inclinations lying within -30° and +30° were rejected. The careful selection procedure finally resulted in a set of 70 ChRM directions providing reliable

paleomagnetic polarity determination at regular intervals from ca. 66 to 290-meter depths of the core. Polarity zones were then defined by at least two consecutive paleomagnetic sites bearing the same polarity such that isolated points were further discarded for the definition of polarity zones.

By considering the above magnetic analysis results 6 reliable polarity magnetozones including 3 normal (N1 to N3) and 3 reversed (R0 to R2) were identified in the section (Fig. S3). As a starting point for the correlation of our paleomagnetic

results with the Geomagnetic Polarity Time Scale (GPTS, Vandenberghe et al., 2012; Ogg et al., 2016) we considered initial dating from fossil pollen biostratigraphy providing independent constraints. Intervals in the core were assigned depositional age ranges based on palynology, in particular the EOB is bracketed between 195.08 and 205.99 m depth (see Sect. 2.2, Table S2, Fig. 2). This interval is comprised near the top of the long reversed polarity zone R1. Above, the relatively short magnetozone N1 overlain by the long R0 provides the expected correlation to the distinctive Chron sequence C13r-C13n-

C12r with the EOB near the top of C13r. The stratigraphic position of the Eocene–Oligocene boundary (EOB) is fixed by assuming that the EOB is situated 86% up in Chron C13r (Luterbacher et al., 2004). According to the above correlation, the two short N2 and R2 magnetozones below would clearly correspond to Chrons C15n and C15r respectively. Below magnetozone R2, the short N3 may consequently correspond to Chron C16n.1n (Fig. 2). Below magnetozone N3, weak paleomagnetic signal with unreliable demagnetization behavior prevented establishing a clear magnetostratigraphic pattern.

### 4.2 Natural Gamma-ray (NGR) variations

NGR values in the studied 65.5-406 m interval vary from ca 50 API to ca 100 API, sometimes reaching greater values but always < 200 API. NGR variations show strongly cyclic patterns with multiple wavelengths and variable amplitudes. In particular, a ~20 m thick cyclicity is prominent throughout the core, on which are superimposed high-frequency cycles (Figs. S4 and S5). Stronger amplitudes are encountered within the Priabonian Stage, especially in its lower part (Fig. S5). The

Lower Rupelian sediments are also characterized by cyclic variations, readily identifiable (Fig. S4), but with lower amplitudes compared to the lower Priabonian interval.





The environmental significance of the NGR variations can be determined based on previous mineralogical and sedimentological results from the core (Tramoy et al., 2016; Ghirardi, 2016). XRD analyses were performed on the various facies along the core and on both bulk rock and the clay fraction (Tramoy et al., 2016). The clay minerals are largely

dominated by kaolinite throughout the core but smectite suddenly appears at ~199.5 m depth with large variations, sometimes in greater proportion than kaolinite (Tramoy et al., 2016). However, this abrupt change in clay minerals assemblage does not affect the NGR periodic signal. This is consistent with a >90% dominated kaolinitic-smectitic fraction that usually does not affect NGR (Hesselbo, 1996). Illite containing potassium is more commonly associated with NGR signal. However, no correlation could be found between illite and NGR variations, probably because, although present, illite

varies little through the core (Tramoy et al., 2016).

In contrast, NGR are clearly anticorrelated to TOC values (Ghirardi, 2016). Although organic matter (OM) is well known for absorbing uranium from marine water in reducing oceanic settings, such anticorrelation (maximum TOC corresponding to minimum NGR) is not uncommon in lake settings and may attest for a diluting process of a U-rich clastic component in the sediment (e.g., Lüning and Kolonic, 2003). In any case, lower NGR intervals corresponding to higher TOC

with enhanced organic matter productivity/preservation conditions are therefore interpreted to reflect protracted wetter periods and vice versa.

### 4.3 Time-series analysis

Power spectrum of the upper interval I1 (Fig. 3A,B) shows three significant (reaching the 99% Confidence Level, CL) low-frequency peaks centred on 45.4, 18.2 and 10.8 m. At higher frequencies, we note a multitude of significant (above 95% CL)

peaks of wavelengths varying from 0.75 to 3.1 m. Power spectrum of the the lower interval I2 (Fig. 3C,D) shows four significant (above 99% CL) low-frequency peaks of 65.7, 22.6, 10.02 and 6.81 m. At higher frequencies, we note a distinct peak centred on 1.01 m, and several other significant (above 95% CL) peaks of wavelengths ranging from 1.67 to 4.67 m.

Filtering, smoothing and visual inspection (Fig. 3A,C) point to a prominent, continuous cyclicity detected by the ~20-m peak (the mean of 18.2 m in Fig. 3B and 22.6 m in Fig. 3D). Visual inspection of well-expressed ~20-m cycles allows

the identification of about twenty high-frequency ~1-meter cycles within almost each of the ~20-m cycle (Figs. S4 and S5). This is consistent with the ~20-m cycles corresponding to 405 kyr eccentricity cycles, and the high-frequency ~1-meter cycles to precession cycles. This cyclostratigraphic interpretation is further substantiated when applying previous timescales (e.g., Vandenberghe et al., 2012) to the recognized polarity chrons in CDB1 Core (Sect. 4.1).

Calibrating the ~20-m cyclicity to a single 405 kyr periodicity results in temporal spectral peaks (Figs. 4 and

S7A,B) sharing some similarities with the astronomical periods (Fig. S7C). In the interval I1, the precession cycle band (Fig. S7A) is characterized by periods ranging from 14 to 22 kyr, the obliquity cycle band between 25 and 54 kyr, and the short eccentricity is possibly detected by the two peaks of 72 and 92 kyr. Within the interval I2, precession corresponds to 17.6 to 19.61 ky, the obliquity to 31.15 to 58.6 kyr, and the short eccentricity possibly to the three peaks at 72, 81 and 134 kyr. We also note two strongly significant peaks in depth and time domains (Fig. S6): the ~10 m peak tuned to ~200 kyr, and the ~55



m peak tuned to ~1020 kyr. The former matches a minor eccentricity term ($2g_2–2g_5$). The latter may correspond to g1–g5 eccentricity term (e.g., Laskar et al., 2004; Abels et al., 2010). We further discussed such cyclicity in Sect. 4.4.

In summary, spectral analysis of the 405 kyr tuned NGR time series shows consistent precession, obliquity, and short eccentricity cycle bands despite potentially missing a few precession cycles by the intrinsic lake responses (e.g., Wang et al., 2020). Additionally, tuning to the 21 kyr mean precession period provides periods of 406 kyr and ~1 Myr for g2-g5

and g1-g5 respectively (Sect. 4.4), thus pointing to the completeness of CDB1 record.

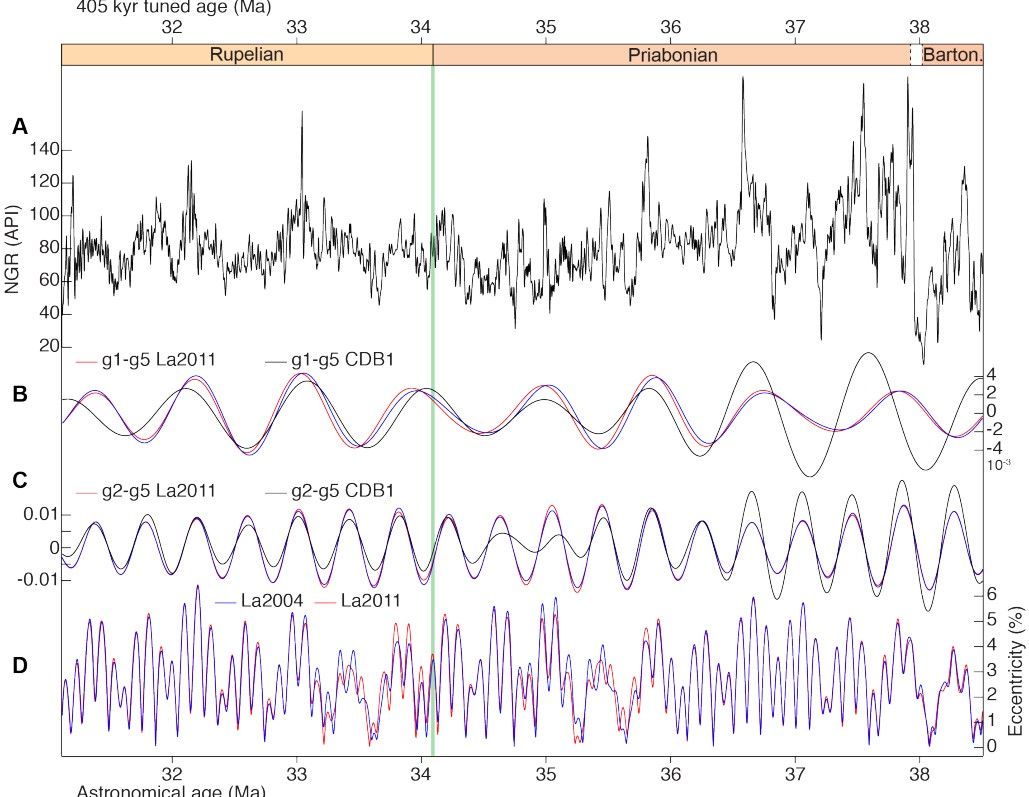

**Figure 4: Astronomical calibration of CDB1 core, along with filtering at 405 kyr (g2–g5) and ~1 Myr (g1–g5) bands. (A) NGR data**
**are tuned to a single 405 kyr sinusoid function, and then anchored at the Eocene-Oligocene boundary at 34.10 Ma. The 34.10 Ma is the age showing the best phase relationship between the tuned NGR and La2004 orbital eccentricity curve (Laskar et al., 2004) at the 405 kyr cyclicity (Fig. S9, see Sect. 4.1 for detail). (B) Bandpass filtering at ~1 Myr band. (C) Bandpass filtering at 400 kyr band. (D) Raw La2004 and La2011 orbital eccentricity data (Laskar et al., 2004, 2011).**





### 4.4 Astronomical calibration of the CDB1 core


The 405 kyr tuning of the NGR data provides a total duration of CDB1 core (interval from 65.5 m to 406 m) of ~7.6 Myr. Durations of the recognized Chrons C12n through C16n.1n, are comparable to estimates from previous studies (Sect. 4.5). Interestingly, we have noted within this time interval the same number of 405 kyr eccentricity cycles, as in deep-sea sequences (Pälike et al., 2006; Westerhold et al., 2014). To make the comparison easier we followed the same numbering

scheme of the 405 kyr eccentricity (Fig. 2), where eccentricity maxima are counted back in time (Hinnov and Hilgen, 2012). In CDB1 Core, Chron C12r through Chron C15r span cycles 78 to 88, exactly the same result as that obtained from the eastern equatorial Pacific, Site 1218 (Pälike et al., 2006), and from the Pacific Equatorial Age Transect (PEAT, IODP Exp. 320/321, Westerhold et al., 2014). Therefore, most of the 405 kyr eccentricity cycles recorded in CDB1 core seem to be complete (consisting of about 20 precession cycles, see Fig. S4). Because the precession frequency is not defined with

precision for the studied time interval (Laskar et al., 2004), our strategy for the astronomical calibration of the core was mainly based on the 405 kyr (first-order) eccentricity cycles. Then, because of the strong expression of the precession signal, a 21 kyr tuning was performed to refine the 405 kyr calibration of polarity chrons (Sect. 4.5), but only on intervals overlapping parts of 405 kyr cycles. These intervals concern cycles 83, 84, 87, 88 and 89, which seem to be complete. Short eccentricity cycles are not as continuously expressed as the 405 kyr cycles, possibly because the obliquity played a role in

obliterating precession modulation. Therefore, short eccentricity cycles were not used for orbital tuning.

The 405 kyr eccentricity tuning of CDB1 core calibrates the longest cyclicity (~50 m thick) to a period of ~1 Myr (Figs. 4 and 5), supporting the expression of g1–g5 in continental environments. In particular, detailed cyclostratigraphic study of middle Miocene fluvial to lacustrine sedimentary sequences in northeastern Madrid Basin (Spain) detected with high fidelity 405 kyr (g2–g5), 0.97 Myr (g1–g5) and 2.4 Myr (g4–g3) eccentricity cycles (Abels et al., 2010).

It seems that the expression of certain eccentricity terms, especially g1–g5, arises from strong imprint of some specific precession components. In particular, longer precession components are susceptible to be better documented than the shorter ones, especially in the context of lower sedimentation rates (Boulila et al., 2008). The two strongest (largest) precession components are p+g5 (23.64 kyr) and p+g2 (22.31 kyr), together with a third long component p+g1 (23.07 kyr), would produce eccentricity modulation terms of three combinations of g2–g5 (405 kyr), g1–g5 (0.97 Myr) and g2–g1 (0.69

Myr) periods (Fig. 5). The CDB1 core documents g2–g5 and g1–g5 eccentricity terms with the greatest amplitudes (Fig. 6), pointing to the importance of these two periodic terms in continental lacustrine records. However, the CDB1 record does not document the 1.2 Myr obliquity nor the 2.4 Myr eccentricity cycles (Abels et al., 2010; Wang et al., 2020).

In addition, we have noticed that sedimentary facies evolution (Bauer et al., 2016) sometimes match the g2–g5 and sometimes the g1–g5 eccentricity terms, supporting the idea of sensitivity of lake level to longer cyclicities (Wang et al.,

2020). Thus, we suggest that the paleolake level may have behaved as a lowpass filter for orbital forcing. Nevertheless, the expression of g1–g5 eccentricity in lacustrine record remains to be investigated. Future cyclostratigraphic studies should





focus on the expression of three specific eccentricity components g2–g5, g1–g5 and g2–g1 in highly resolved continental records.

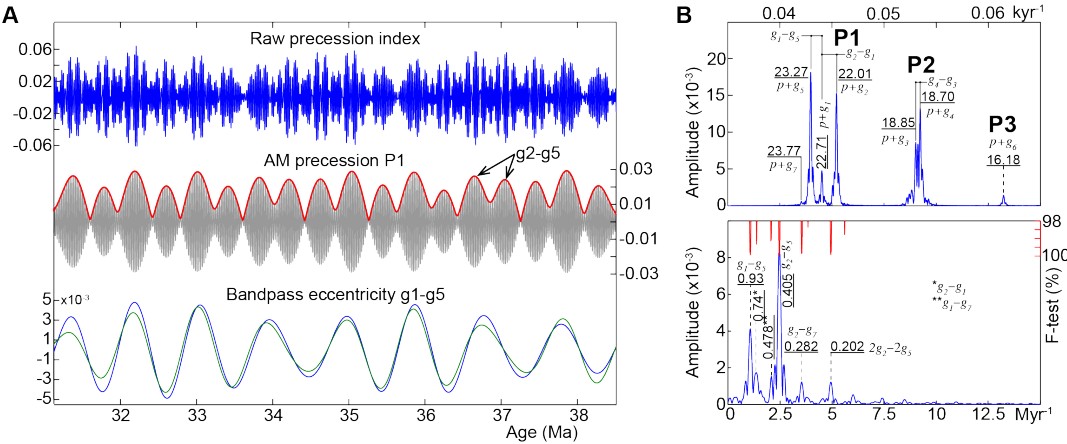


**Figure 5:** Amplitude modulation (AM) of the precession index band P1. (A) La2004 precession index (Laskar et al., 2004), AM of precession P1 band (cutoff frequencies: 0.044 ±0.002 cycles/kyr), and bandpass filtering of g1-g5 eccentricity component (cutoff frequencies: 0.00115 ±0.0003 cycles/kyr) from AM precession P1 (blue curve) and from the raw eccentricity (green curve). (B) 2π-

MTM power spectra of the raw precession index (upper panel) and AM precession P1 (lower panel).





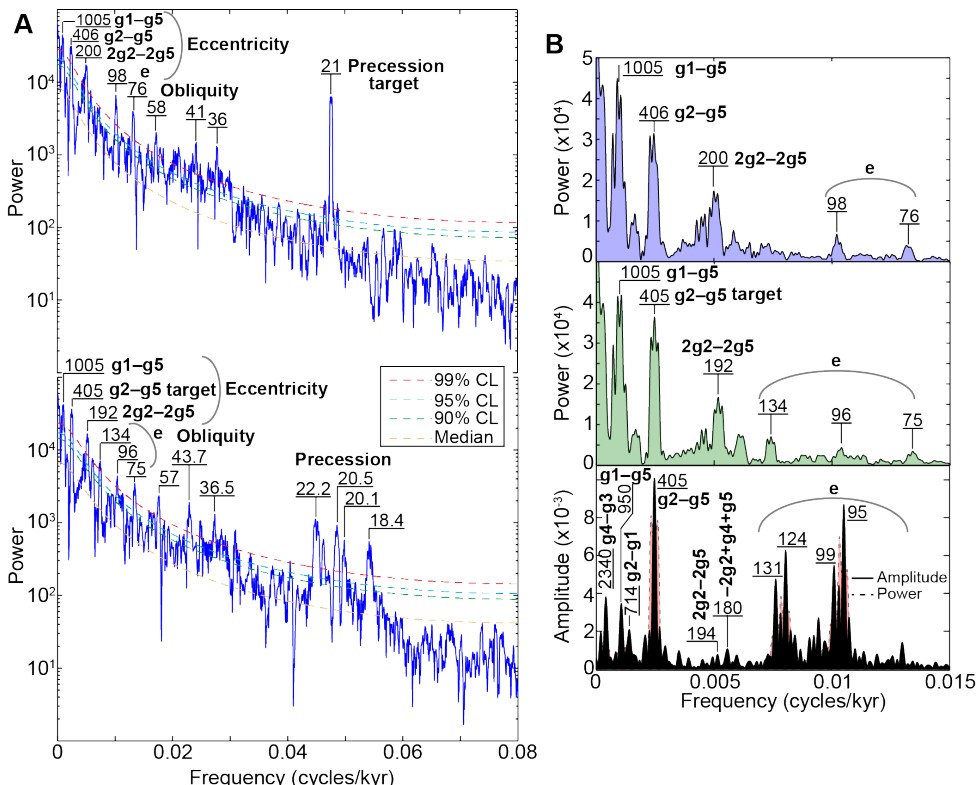

**Figure 6: 2π-MTM power spectra of the whole tuned raw NGR data. (A) (Upper) spectrum of the 21 kyr precession (Fig. S7C) tuned NGR, (lower) spectrum of the 405 kyr tuned NGR. (B) Upper and middle spectra are the same as in 'A' but for the interval 0 to 0.015 cycles/kyr and in a linear scale power axis, to highlight the low-frequency portion of the spectra. Lower spectrum of La2004 eccentricity over the equivalent time interval (31.110 to 38.501 Ma). Secular frequency for each eccentricity component is given, except for the short eccentricity (e) which has several components (e.g., Laskar et al., 2004).**


## 4.5 Durations of C12r through C16n.1n

Below we compare CDB1 timescale with previous astronomical timescales in terms of calibrated durations of Chrons C12r through C16n.1n (Table 1). A recent study of Sahy et al. (2020) has shown difficulties in the quantification of ages of Chrons C12n through C16n.2n, even when integrating astronomical and radioisotopic dating. Accordingly, we have preferred to

discuss durations of chrons rather than their ages (Table 1).

Duration of Chron C12r is very close to estimates derived from orbital tuning of ODP Site 1218 (Pälike et al., 2006) and PEAT sites (Westerhold et al., 2014), but longer than other estimates, e.g., 0.146 Myr longer than in CK95 (Cande and





Kent, 1995) and GTS2012 (Vandenberghe et al., 2012). Both studies at Site 1218 and PEAT sites used well expressed 405 kyr orbital eccentricity cycles in physical and chemical proxies to infer durations of 2.211 and 2.237±0.018 Myr respectively, which are very close to 2.256 Myr assessed here in CDB1 Core.

Duration of Chron C13n in CDB1 Core is estimated to 0.535 Myr, which is consistent with recent estimates, 0.550 Myr (Vandenberghe et al., 2012) and 0.512±0.010 Myr (Westerhold et al., 2014). These durations are, however, longer than those proposed by older studies, which yielded durations of 0.472, 0.473 and 0.487 Myr (Table 1). In fact, the duration of 0.473 Myr proposed by Pälike et al. (2006) using orbital tuning of Site 1218 was revised by Westerhold et al. (2014) using the PEAT data, and additional ODP sites including Site 1218. Therefore, Chron C13n would have a duration between ca. 0.50 and 0.55 Myr.

The duration of Chron C13r in CDB1 Core is assessed at 1.377±0.021 Myr, which is similar to estimates from Site 1218 (Pälike et al., 2006), and PEAT Sites (Westerhold et al., 2014), but much shorter than durations proposed in other timescales, e.g., CK95 and GTS2004 (Table 1).

The duration of Chron C15n assessed here at 0.304 ±0.010 Myr is identical to GTS2012 (Vandenberghe et al., 2012), but longer than other duration estimates. Duration of Chron C15r in CDB1 Core is estimated at 0.325±0.010 Myr, which is shorter than those inferred from previous GPTS, but longer than previous ATS estimates (Pälike et al., 2006; Westerhold et al., 2014). In contrast, the duration of Chron C16n.1n estimated in CDB1 Core at 0.105 Myr is much shorter than all previous timescales.

In summary, durations of Chrons C12r, C13n and C13r are going towards a stable orbitally tuned timescale, where estimates from the continental CDB1 record are very consistent with previous estimates inferred from orbital tuning of deep-sea sequences. However, durations of Chrons C15n, C15r and C16n.1n, as mentioned in a previous study (Westerhold et al., 2014), still suffer from uncertainty. The CDB1 Core provides additional constraints on durations of C15n and C15r, but duration of C16n.1n is too short (reduced) to be retained.

| Polarity Chron | CK95 | GPTS2004 | HP2006 | GPTS2012 | TW2014 | WRG | UMB | CDB-1 Core (this study) |
|---|---|---|---|---|---|---|---|---|
| C12n | 0.46 | 0.489 | 0.404 | 0.44 | | | | |
| C12r | 2.119 | 2.150 | 2.211 | 2.12 | 2.237 ±0.018 | 2.01 ±0.13 | 1.87 ±0.07 | 2.266 |
| C13n | 0.487 | 0.472 | 0.473 | 0.55 | 0.512 ±0.010 | 0.4 ±0.13 | 0.64 ±0.09 | 0.535 ±0.010 |
| C13r | 1.110 | 1.044 | 1.421 | 1.29 | 1.376 ±0.041 | 1.14 ±0.11 | 1.17 ±0.09 | 1.377 ±0.021 |
| C15n | 0.285 | 0.261 | 0.128 | 0.30 | 0.234 ±0.049 | 0.16 ±0.10 | 0.2 ±0.09 | 0.304 ±0.010 |
| C15r | 0.403 | 0.361 | 0.074 | 0.41 | 0.244 ±0.031 | 0.38 ±0.08 | 0.32 ±0.05 | 0.325 ±0.010 |
| C16n.1n | 0.183 | 0.163 | 0.226 | 0.19 | 0.138 ±0.036 | 0.26 ±0.05 | 0.07 ±0.03 | 0.105 |

**Table 1: Durations of Chrons C12r through C16n.1n inferred from CDB1 core cyclostratigraphy and comparison with previous timescales. Geomagnetic Polarity Time Scales: CK95 (Cande and Kent, 1995), GPTS2004 (Ogg and Smith, 2004) and GPTS2012 (Vandenberghe et al., 2012), Astronomical Time Sacle (ATS): HP2006 from ODP Site 1218 (Pälike et al., 2006), TW2014 from ODP PEAT sites (Pacific Equatorial Age Transect, Exp. 320) (Westerhold et al., 2014). Magnetostratigraphic and radioisotopic**



**dating: WRG and UMB are White River Group and Umbria-Marche Basin respectively (Sahy et al., 2020). Note that durations of all these chrons in GTS2020 (Gradstein et al., 2020) are from Westerhold et al. (2014).**


## 5 Discussion

### 5.1 The age of the Eocene-Oligocene boundary

The late Eocene-early Oligocene timescale still lacks precision, and the age of Eocene–Oligocene boundary (EOB) is not yet fixed, thus several estimates ranging from 33.714 Ma to 34.1 Ma have been proposed (Table S4, see Hilgen and Kuiper,
2009 for a review). In particular, the late Eocene timescale has proven difficult to tune in several ocean drilling programs because of the shallow position of the calcite compensation depth (CCD) preventing deposition of carbonate-rich sequences suitable for application of integrated bio-cyclostratigraphy and stable isotopic stratigraphy (Pälike and Hilgen, 2008; Boulila et al., 2018). Although recent efforts have been made to tune this time interval from the Pacific Equatorial Age Transect (PEAT, IODP Exp. 320/321, Pälike et al., 2010), more precision is still needed to refine this key part of the Paleogene–
Neogene (Westerhold et al., 2014).

Since astronomical calibration of the middle-late Eocene is not well-constrained in GTS2012, Vandenberghe et al. (2012) preferred to calculate ages for Chron boundaries using the synthetic marine magnetic anomalies profile of Cande and Kent (1995) and an interpolation between the nearest older and younger astronomically dated Chron boundary (i.e., C21n(o) at 47.8 Ma, and C13n(o) at 33.705 Ma). This combined Paleogene age model resulted in an age estimate of 33.9 Ma for the
EOB. A more recent study of orbital tuning of middle Eocene to early Oligocene using mainly data from the PEAT sites arrived at a close age of 33.89 Ma for the EOB (Westerhold et al., 2014). A younger 33.714 Ma age has proposed by Jovane et al. (2006) on the basis of tuning the Rupelian GSSP of Massignano, and is also close to one option of van Mourik et al. (2006) from the Massicore, drilled ~100 m south of the Massignano section. In addition, van Mourik et al. (2006) provided another older age option of 34.1 Ma. A more recent study (Sahy et al., 2017) re-evaluated the late Eocene through
Oligocene timescale using integrated radioisotopic and cyclostratigraphic data to reach an age of 34.09 ±0.08 Ma for the EOB.

Our cross-correlations between the 405 kyr CDB1 timescale anchored at all these previously proposed EOB ages (Table S4) and the Earth's orbital eccentricity sheds some light on these controversies. Results of cross-correlations between NGR and astronomical variations are summarized in Fig. S9. The 33.90 and 33.89 Ma ages (and less the 33.95 Ma) of the
Eocene–Oligocene boundary (EOB) both show opposite phase relations between the NGR and eccentricity but the 33.714 Ma and 34.1 Ma EOB ages are approximately in phase. The other proposed ages result in NGR and eccentricity neither in phase or in opposite phase. We deduce that the optimal EOB ages in terms of phase relationship between the NGR and astronomical signal are either 33.714 and 34.10 Ma that are in phase or 33.90 and 33.89 Ma that are in opposite phase (Fig. S9). We can further refine this selection by considering 405 kyr eccentricity cycle maxima as responsible for stronger
amplitude precession and insolation cycles, and thus intensified seasonality with longer drier seasons. Based on pollen





assemblage proxy, the prevailing climatic conditions within the studied interval (or at least from 66.85 to 265.90 m) were dry (Bauer et al., 2016). Thus, increasing range between dry- and wet-season precipitation over land and its effect on vegetation primary productivity is likely, leading in general to longer and more intense dry seasons (e.g., Murray-Tortarolo et al., 2016). Following the anticorrelation of NGR and TOC and the hypothesis of cyclically diluted NGR by the organic matter (Sect.

4.2), such astroclimatic link would favor higher NGR but lower TOC values during insolation maxima, related to reduced organic matter productivity/preservation conditions during protracted drier seasons. Accordingly, increased values of NGR would reflect higher values in the orbital eccentricity, i.e., NGR and eccentricity would be in phase and thus favour the 33.714 Ma and 34.1 Ma EOB ages.

**5.2 Environmental changes across the Eocene-Oligocene transition**

Based on our age control we can investigate the precise timing of previously observed environmental changes at the CDB1 core (Tramoy et al., 2016; Fig. 7) and correlate them to the well-established EOT evolution in the marine record and associated climate interpretations. In marine records, the Eocene–Oligocene climate transition (EOT) is described in two major steps (Coxall et al., 2005; Katz et al., 2008). The first (older) step (or EOT-1, Katz et al., 2008) occuring in the upper part of Chron C13r. The second (younger) climatic step (or Oi-1, Miller et al., 1991, Zachos et al., 1996), which represents

the major shift in $\delta^{18}$O, is recorded around the base of Chron C13n. The first step is associated with atmospheric cooling ascribed to declining atmospheric $CO_2$ and leading to the second step marking the onset of the major Antarctic glaciation (DeConto and Pollard, 2003; Pearson et al., 2009; Ladant et al., 2014). The Oi-1 glacial event could be divided into two sub-steps, Oi-1a and Oi-1b (Zachos et al., 1996; Coxall and Wilson, 2011). High-resolution $\delta^{18}$O data from eastern equatorial Pacific (Coxall and Wilson, 2011) compared to other $\delta^{18}$O data mainly from the Southern Ocean (e.g., Zachos et al., 1996)

further confirm the two-step structure of the EOT, but support in addition an EOT precursor ice growth event (Sect. 5.3) within Chron C13r (see Katz et al., 2008; Peters et al., 2010; Pusz et al., 2011).

The CDB1 core exhibits significant changes in the sedimentary facies around the EOT (Bauer et al., 2016). In particular, two prominent sedimentary facies changes occuring slightly below and above the EOB respectively. We estimate the duration of the interval between the two facies changes to 0.268 Myr. This interval comprises thirteen well-expressed

precession cycles, further supporting a duration around 0.27 Myr. This compares nicely with marine records of steps EOT-1 and Oi-1 below and above the EOB, separated by about 0.25 Myr (Katz et al., 2008; Coxall and Wilson, 2011). Therefore, these two sedimentary changes in CDB1 core most likely reflect the two major climatic steps EOT-1 and Oi-1. We note that Oi-1a, Oi-1b and the EOT precursor event do not appear to be expressed by lithological changes at CDB1.

The first step, below the EOB (205.5 m) (Fig. 7), is expressed as a gradual but rapid decrease in TOC (from black,

organic-rich clays, to light grey, whitish clays in a few decimeters). The second change above the EOB, which occurs at depth 195.26 m, is expressed by a sharp contact between massive, clotted greenish clay (palustrine facies) and overlying brownish, organic laminated clays (deep lacustrine facies). Previously acquired environmental proxies across the EOT at



CDB1 show a TOC decrease, the appearance of smectite at the expense of kaolinite and a significant $\delta^{15}$N increase (Tramoy et al., 2016). The EOT is also marked by a strong development of conifers and the development of Ephedraceae and

Umbelliferae (Bauer et al., 2016). Together these proxies have been interpreted as a rapid transition from humid and warm to drier and cooler climatic conditions with increased seasonality. In detail, the first step corresponds to the onset of the $\delta^{15}$N increase, a sharp TOC decrease and coeval increase in quartz contents from ~13 to ~49 % suggesting respectively, drying, dwindling vegetation cover and enhanced erosion in the catchment. At the second step, $\delta^{15}$N continue to increase but a step cannot be deciphered due to insufficient sampling resolution (note different position of the second step in Tramoy et al.,

2016). However, a step is clearly picked up by quartz content showing another major rapid increase from ~20 to ~70%, suggesting together with $\delta^{15}$N, further drying and catchment erosion while TOC remains low (~0.5 to ~3 %), and later recovers (~12 %) (Fig. 7). Increased sediment input and drainage perturbation is also suggested by increased accumulation rates between the two steps before reaching steady deep lacustrine facies. This highlights the instability during this transition between climate states. Interestingly, the smectite signal seems delayed, appearing 6 meters (147 kyr) above the first step

and becoming dominant at the expense of kaolinite only 15 meters (357 kyr) after the second step which may reflect lag time in clay formation, transport and accumulation.

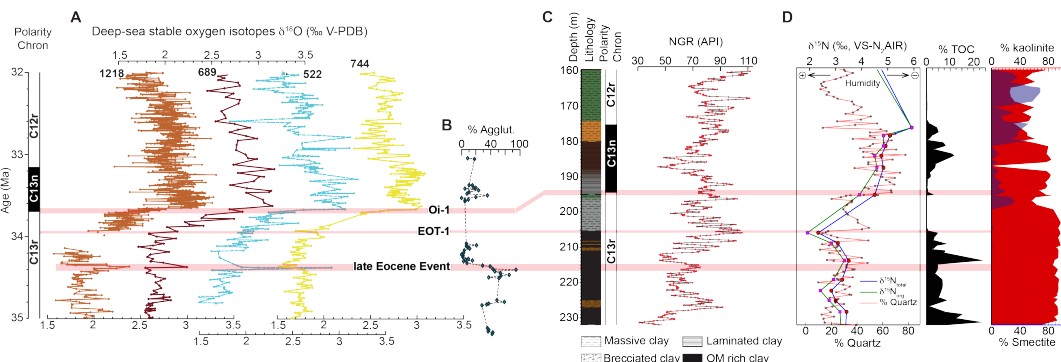

**Figure 7: Correlation of the main changes in the sedimentary facies in CDB1 Core around the EOT and the main climatic (cooling-glacial) events recognized in deep-sea records (DSDP Site 522, ODP Sites 689, 744 and 1258). (A) Deep-sea δ18O data from multiple sites (indicated, Site 1218: Coxall and Wilson, 2011; Site 689: Diester-Haas and Zahn, 1996; Sites 522 and 744: Zachos et al., 1996) document successive glacial events: "late Eocene Event", EOT-1 and Oi-1 related to the Antarctic Ice Sheet (Coxall and Wilson, 2011). (B) Agglutinated benthic foraminifera data at ODP Site 647 (Kaminski and Ortiz, 2014) against the age**

**model of Coxall et al. (2018). (C) Lithostratigraphy, magnetostratigraphy and NGR data around the Eocene-Oligocene boundary in CDB1 core. (D) Terrestrial-derived organic nitrogen isotope record (δ15Norg), quartz content, total organic carbon (TOC) content, kaolinite and smectite contents from Tramoy et al. (2016).**



In summary, CDB1 records atmospheric cooling, drying and increased seasonality through the EOT as previously documented, with local variations, in continental records worldwide (Pound and Salzmann, 2017; Toumoulin et al., 2021 and references therein). At the CDB1, drying may simply relate to decreased intensity of the hydrological cycle due to global temperature and $CO_2$ decrease as expected by climate models (e.g., Ladant et al., 2014). Alternatively, the North Atlantic coastal location of the CDB1 site, opens the possibility that observed environmental changes relate to rerouting of oceanic

currents leading to increased latitudinal gradients associated with the onset of the ACC and/or to the onset of a proto-Atlantic Meridional Oceanic Circulation (AMOC; see Sect. 5.3; Abelson and Erez, 2017; Elsworth et al., 2017; Coxall et al., 2018)

        In any case, the two recognized environmental step changes concur with the global impact of the EOT-1 and Oi-1 events on terrestrial environments expressed via the hydrological cycle (e.g. Xiao et al., 2010; Page et al., 2019; Licht et al., 2020). Drying expressed by $\delta^{15}N$ seems to occur through both steps but vegetation appears mostly affected by EOT-1.

Further CDB1 proxy (in particular temperature) analyses at higher resolution will be required to assess changes at specific steps to infer their potential forcing mechanisms. However, the EOT cannot be simplified as a suite of discrete steps because it is clearly imprinted by orbitally paced variability (Zachos et al., 2001b; Coxall et al., 2005, 2011) that also provides insight on forcing mechanism as developed below.

**5.3 Obliquity forcing preconditionnd the EOT**

Visual inspection of the NGR variations points to prominent obliquity-scale cycles (with very weak or no precession cycles) occurring in the upper part of Chron C13r, within the 405 kyr cycle no. 85 (Fig. 8). The dominance of the obliquity starts ca. 216.5 m, and ends ca. 205.5 m. Evolutive harmonic analysis of the interval spanning the 405 kyr cycles 84, 85 and 86 (186-232 m interval) highlights a switch from obliquity-dominant to precession-eccentricity-dominant cycles at EOT-1. In detail, from depth 232 to ~216.5 m, NGR variations record precession, eccentricity and strong obliquity. From 216.5 to 205.5 m the

precession signal is weak, and the obliquity dominates (see also Fig. S10). From 205.5 to 186 m, the precession is well expressed, and the eccentricity is stronger than the obliquity. The time span between the onset (beginning) of obliquity-dominant cycles and the EOT-1 equivalent level (Sect. 5.2) is 0.280 Myr, which is in the same order than the 0.3 Myr proposed by Katz et al. (2008) between the EOT precursor Glacial Event  (LEGE) and the EOT-1 (Figs. 7 and 8). Accordingly, we suggest that the interval that documents the dominance of obliquity cycles corresponds to the LEGE - EOT-

1 interval. This interval shows a gradual cooling confirmed by Peters et al. (2010), Pusz et al. (2011), and Coxall and Wilson (2011) on the basis of deep-sea data. In addition, several studies have recognized obliquity forcing in the latest Eocene terrestrial and marine records and argued it may relate to the presence of polar ice preconditionnning the EOT (e.g. Abels et al., 2011; Jovane et al., 2006; Brown et al., 2009). Results from the CDB1 core lend support to the idea of obliquity forcing before EOT-1 but its cause and potential significance for the EOT remains to be determined amongst competitive

mechanisms.

        On the one hand, the global effect of large Antarctic ice-volume variations dominated by obliquity before its continental scale coalescence, may have modified the continental hydrological cycle and eustasy (e.g., deConto and Pollard,



**Climate**
**of the Past**
Discussions

2003; Ladant et al., 2014). After the ice-sheet formation at Oi-1, less important changes in ice volume would have ended obliquity dominance over precession-eccentricity forcing. This supposes that polar ice had formed earlier than EOT-1 in

response to a more gradual $CO_2$ decrease (e.g., Pound and Salzmann, 2017). Note these orbitally-forced eustatic variations during early opening of the shallow Drake passage may have also enabled incipient periodic ACC installation.

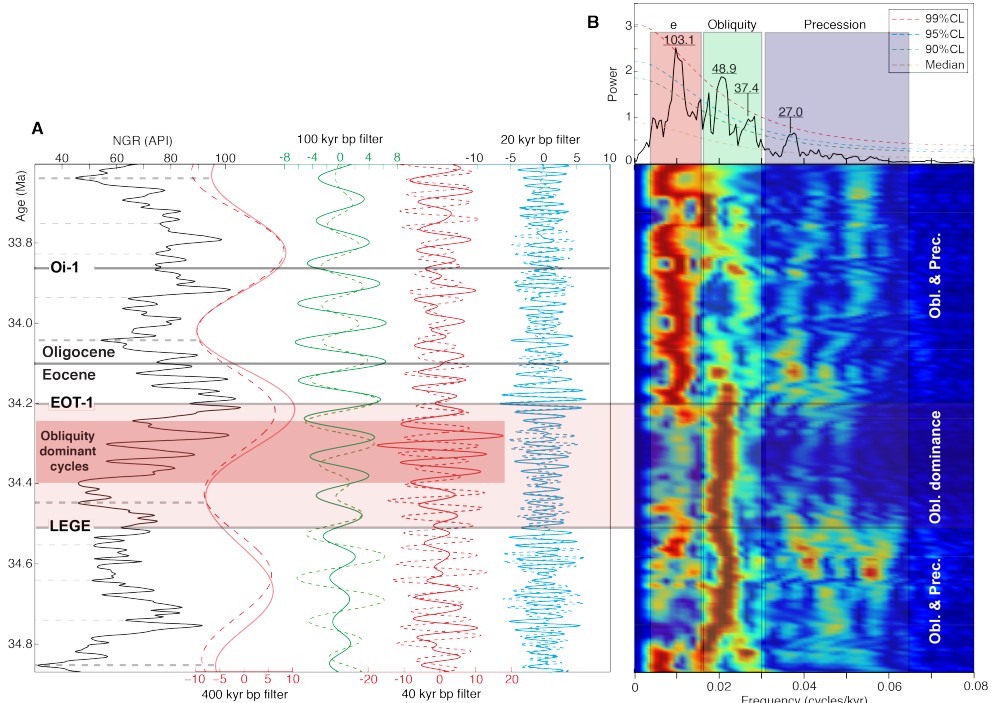

**Figure 8: Time-series analysis focused on the Eocene–Oligocene transition (EOT) showing the switch from obliquity to precession-eccentricity. (A) Tuned NGR time series for the interval from 186 to 232 m, anchored at the EOB age of 34.10 Ma, with bandpass filtering at the 100 and 400 kyr eccentricity, 40 kyr obliquity and 20 kyr precession bands (solid curves for NGR cycles and dashed curves for La2004 astronomical cycles). Shaded area indicates strong obliquity cycles. (B) Power spectrum and amplitude spectrogram of the detrended (400 kyr cycles removed) tuned NGR, with the interpreted astronomical cycles for the 100 kyr**
**eccentricity (e), obliquity and precession. Periods in the spectrum are in kiloyears. Note the striking switch from obliquity-dominant to precession-eccentricity cycles at the Eocene-Oligocene Transition event 1 (EOT-1). The interval from the late Eocene glacial event (LEGE) to the EOT-1 records very weak precession, especially at the dark pink-shaded area.**

On the other hand, the pre-EOT-1 obliquity interval may relate to the onset of North Atlantic Deep Water (NADW)
production, driven either by the opening of the Drake Passage (Abelson and Erez, 2017; Elsworth et al., 2017) or tectonic adjustments allowing southward flow of cold and nutrient laden Arctic water fuelling a proto-AMOC (Coxall et al., 2018).



The latter hypothesis would have led to northward heat transport responsible for Antarctic cooling preconditioning the EOT. Multi-proxy study of North Atlantic sites suggests North Atlantic deep water formation starting at the LEGE and ending at EOT-1 (e.g., Kaminski and Ortiz, 2014; Coxall et al., 2018), precisely like the obliquity dominant interval at CDB1.

Specifically, a remarkable decline in agglutinated benthic foraminifera in the southern Labrador Sea was ascribed to increased seawater cell convection in relation with the formation of deep waters (Fig. 7B, Kaminski and Ortiz, 2014; onset of regime 3 in Coxall et al., 2018). This abrupt biotic turnover shift is ~630 kyr older than the EOB (Coxall et al., 2018 their Fig. 3F). In the continental CDB1 core, obliquity started to dominate ~650 kyr before the EOB (Fig. 8). This remarkably coincident timing leads us to suggest the obliquity-dominated CDB1 interval support the NADW formation hypothesis. This

is substantiated by (1) the CDB1 location directly affected by NADW, (2) the NADW sensitivity to obliquity and (3) the expected westerly moisture source of CDB1 governed by the NADW (e.g., Abelson and Erez, 2017). Nevertheless, this hypothesis still needs further investigations from other terrestrial sites, deep-sea data records and climate modeling.

## 6 Conclusions

A ~7.6 million year-long lacustrine gamma-ray (NGR) record spanning the late Eocene (latest Bartonian to Priabonian) to

the earliest Oligocene (early Rupelian) from a drill-core (CDB1) in the Rennes Basin (Northwestern France) documents with high fidelity the astronomical frequencies (precession, obliquity and eccentricity). The stable 405 kyr eccentricity is continuously expressed, and thus used to astronomically calibrate this record.

The most prominent NGR cyclicity is calibrated to a period of ~1 Myr matching g1–g5 eccentricity term. Such cyclicity has been recorded in several continental records, pointing to its strong expression in continental depositional

environments. The mechanisms transferring such cyclicity into the sedimentary lacustrine record remain elusive. Nevertheless, we suggest that the paleolake level may have exerted a lowpass filtering of the recorded astronomical variations since the low-frequency g1-g5 and sometimes the g2-g5 cyclicity are equally expressed in the sedimentary facies evolution.

Duration estimates of Chrons C12r through C16n.1n from the 405 kyr orbital tuning is consistent with previous

estimates obtained from deep-sea sequences. This attests to the completeness of the CDB1 lacustrine record. Such exceptional terrestrial record motivates us to suggest the CDB1 timescale as a continental reference in a future generation of Paleogene-Neogene timescales.

Assuming that maxima in NGR data reflect maxima in Earth's orbital eccentricity (maxima of insolation) and by anchoring the 405 kyr tuned NGR time series at different available ages of the Eocene–Oligocene boundary (EOB) we have

noted that the 33.714 and 34.10 Ma are the optimal ages providing the best phase fit between the NGR record and orbital eccentricity variations.

Improved calibrated age control enables to identify that the EOT-1 and Oi-1 events are both marked by environmental changes (cooling, drying and enhanced seasonality) recorded in the CDB1 lithology and proxy records. This



support previous claims that these events had global impact on terrestrial environments via the hydrological cycle but controlling mechanism remain elusive.

Detailed orbital study around the Eocene–Oligocene transition (EOT) highlights a pre-EOT interval dominated by obliquity over precession-eccentricity, thus concuring with previous studies reporting similar patterns in terrestrial and oceanic environments, and their possible link with Antarctic glaciation preconditionning. At the end of this interval, the CDB1 core records for the first time a switch from obliquity to precession-eccentricity forcing at around the EOT-1 event.

We thus potentially relate the obliquity-dominated interval to the instable phases of the incipient Antarctic Ice Sheet (AIS), whereas the onset of precession-eccentricity dominated after EOT-1 would relate to a more stable, nearly coalesced AIS. These observations lead us to consider obliquity variations preconditioned the EOT, either directly through AIS modulation or, preferably, via North Atlantic Deep Water formation.

**Data availability**

All new data presented in this paper can be found in the Supplement

**Supplement**

The supplement related to this article is available online at:

**Author contribution**

The manuscript was conceived and planned by SB and GDN, in collaboration with all authors. SB carried out the cyclostratigraphic analyses. GDN conducted the paleomagnetic study. All authors contributed to the interpretations of data and finalization of the manuscript.

**Competing interests**

The authors declare that they have no conflict of interest

**Acknowledgements**

This work originated from a project (CYNERGY Project) funded by the BRGM, the AELB and the ADEME. All the scientific crew is thanked for fruitful discussion.



**Financial support**

SB and BG have been supported by ANR AstroMeso and ERC AstroGeo projects. GDN has been supported by the Marie-
Curie CIG HIRESDAT and ERC MAGIC grants 649081.

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
