# Peer review of "Age and driving mechanisms of the Eocene-Oligocene Transition from astronomical tuning of a lacustrine record (Rennes Basin, France)"

_Climate of the Past, 2021_

## Author Response (AR1)

Dear Editor,

Please find below our detailed responses to the reviewers' comments. The new text is highlighted in pink color, in the marked-up version of the revised manuscript. We have also updated some figures, and added new figures (figure 2B and figure 4) according to reviewers' comments. All of these changes are explicitly mentioned below in the responses to the reviewers' comments, shown in blue text.

Very sincerely,

Slah Boulila

Corresponding co-author

**Comment on cp-2021-46**

**Anonymous Referee #1**

*Referee comment on "Age and driving mechanisms of the Eocene-Oligocene Transition from astronomical tuning of a lacustrine record (Rennes Basin, France)" by Slah Boulila el al., Clim. Past Discuss., https://doi.org/10.5194/cp-2021-46-RC1, 2021*

**General comments**

*The geochronology of the Eocene-Oligocene transition remains controversial due to the lack of stratigraphic records. This study presents an astronomically calibrated magnetostratigraphy for this transition. Well-defined magneto-zones help high-resolution correlation between the studied lacustrine record to the deep-sea cores. Time series analysis further refines the geochronology of this study. This paper worth publication after addressing the following major comments:*

**Specific comments**

1. *This paper aims to investigate whether the cyclic lacustrine deposits are orbitally driven. The analysis, aided with magnetostratigraphic correlation does answer this critical question. However, I would highly recommend the authors considering running statistical tuning methods, either ASM, COCO, or TimeOpt analysis of the gamma-ray data to test the significance level of the null hypothesis of no orbital forcing because the traditional cycle ratio method can generate misleading cycle ratios which lead to misinterpretation. Moreover, the sedimentation rate map is also expected to prove the assumed steady sedimentation rate is robust. The only figure S8 of the evolutive harmonic analysis shows fair results and very unclear implications of sedimentation rate. Therefore, an evolutionary version of ASM, COCO, or TimeOpt would help eliminate this question. Because the original data is unavailable now, so I wouldn't be able to reproduce the results, although the figures provided looks fine.*

**Reply to Comment #1_ cp-2021-46_ Referee #1:**
We agree. As stated by the reviewer, the chronostratigraphic framework from magnetostratigraphic constraints can answer whether the cyclic lacustrine deposits are orbitally-driven. This method is generally used for the Cenozoic cyclostratigraphy. Yet, we agree that statistical approaches to demonstrate the orbital forcing, usually applied to Mesozoic records (without relatively precise age controls), can also provide additional confidence for Cenozoic records.

Accordingly, we now provide statistical methods based on the COCO and the evolutionary COCO analyses in order to test the significance level of the null hypothesis of no orbital forcing, and to estimate the evolution of sedimentation rate throughout the core.

The results are provided in figure 2 for evolutionary COCO results, and in the new figure 3 for the 'single' COCO results. As expected, these results support our previous cyclostratigraphic interpretations based on the preliminary magnetostratigraphic age model and on the manual use of the frequency ratio method.

> 2. *This paper also aims to refine the Paleogene time scale. It is a great pity that recent advances in the Eocene geochronology were not cited and discussed. Key publications include the GTS2020, Westerhold et al. (2020 Science), and Berggren et al., 2018 (http://orca.cf.ac.uk/117311/1/Chapter_2.pdf). These publications presented the latest ages for the studied magneto-zones. And the GTS suggested a 33.9 Ma EOT, which contradicts the 33.7 or 34.1 Ma EOT age in this paper.*

**Reply to Comment #2_ cp-2021-46_ Referee #1:**
We should have better explained that the latest geological timescale GTS2020 (Gradstein et al., 2020; Speijer et al., 2020_The Paleogne Period Chapter) that we cited, and Westerhold et al. (2020) both rely on the ages from Westerhold et al. (2014), based on cyclostratigraphy of IODP Expedition 320 sites. Thus, there is no change in ages and durations of magnetic polarities C12r through C16n.1n between Westerhold et al.'s (2014) study and GTS2020. We note that some of Westerhold et al.'s (2014) results are included in GTS2016 (Ogg et al., 2016) and that the GTS2016 and GTS2020 are cited in our paper. We have tried to clarify these points in the associated parts (see Supplementary Table S4, and details in Section 5.1).

Indeed, we had not discussed sufficiently the differences in EOB ages. We are well aware Westerhold et al. (2014) provided an age of 33.89 Ma for the Eocene/Oligocene boundary (EOB) and that a slightly different age of 33.90 Ma is used in all previous geological timescales GTS2012, GTS2016 and GTS2020. Despite significant advances and the apparent agreement in these proposed ages in astronomically calibrated Cenozoic timescale, it should be clear that controversial ages of the EOB unfortunately still exist (discussed in depth in Hilgen and Kuiper, 2009, see also Sahy et al., 2017). Thus, we have tested this 33.90 Ma age of the EOB retained by the geological timescales together with all previously suggested ages (see Supplementary Table S4, and details in Section 5.1) to investigate the phase relationship between the sedimentary NGR data and the theoretical orbital eccentricity variations.

We had noted in the caption of Table 1 that:

*« Note that durations of all these chrons in GTS2020 (Gradstein et al., 2020) are from Westerhold et al. (2014). ».*
We revised it as follows:
*« Note that durations of all these chrons in GTS2020 (Gradstein et al., 2020) are from Westerhold et al. (2014) and subsequently used in Westerhold et al. (2020). ».*

In addition, to further highlight this note to the reader, we now added some clarifications in the first paragraph of Section 4.5 for the magnetic polarities, and in Section 5.1 for the age of EOB.

We also now cite the interesting paper of Berggren et al. (2018) suggested by the reviewer, focused on chronostratigraphy of planktonic foraminiferal biostratigraphy, and suggested two alternatives for the age of EOB, i.e. 33.70 and 33.90 Ma. These ages are based on previous studies that we extensively discussed in the paper (Section 5.1). In particular, these two proposed ages (and other ages we used) depend on age calibration used on the Fish Canyon Tuff standard, as has been reviewed by Hilgen and Kuiper (2009). This further supports the hypothesis that the age of EOB is not yet resolved.

3. *Lacustrine records usually suffer from missing high-frequency astronomical cycles (obliquity and precession) and pollution from autogenic sedimentary cycles (Hajek and Straub, 2017). Therefore, the claimed 1 m scale precession cycles may be suspicious. I would like to see the argument against this comment.*

**Reply to Comment #3_ cp-2021-46_ Referee #1:**
This is indeed a controversial topic since many lacustrine records across the world and in various geological settings have been shown to provide excellent high-frequency orbital cycles (or even sub-orbital cycles). In our record, we can simply argue against the hypothesis that the 1 m scale precession cycles are suspicious by highlighting the multiple expanded views of the original (raw) NGR variations in the depth domain (provided in the Supplementary materials Figures S4, S5, S11). There, the 1 m wavelength cyclicity is well preserved in the highly resolved NGR data. These clearly show well-expressed 1 m scale cycles defined by several data measurement points. For instance, upon simple visual inspection of figures S4 and S5, 18 to 21 one-m-scale cycles are clearly apparent within each 20 m scale cycle. This can be interpreted as reflecting respectively the precession and the 405 kyr eccentricity periodicities.

This is indicated now in lines 284-288 as follows with references to the associated figures in the supplementary:
*"Visual inspection of well-expressed ~20-m cycles allows the identification of about twenty high-frequency ~1-meter cycles within almost each of the ~20-m cycle (Figs. S4 and S5). This is consistent with the ~20-m cycles corresponding to 405 kyr eccentricity cycles, and the high-frequency ~1-meter cycles to precession cycles".*

4. *All citations in blue are listed, however, citations in black are missed. Make sure all cited publications are listed at the end of the paper.*

**Reply to Comment #4_ cp-2021-46_ Referee #1:**
Good point, we have checked all references cited in black, and added them in the reference list.

**Comment on cp-2021-46**
Anonymous Referee #2

*Referee comment on "Age and driving mechanisms of the Eocene-Oligocene Transition from astronomical tuning of a lacustrine record (Rennes Basin, France)" by Slah Boulila et al., Clim. Past Discuss., https://doi.org/10.5194/cp-2021-46-RC2, 2021 ;*

*#1 This study presents an astronomical time scale from ~31 to ~39 Ma. It is worth of investigation since there are unresolved precise time scale for this period. However, as the RC1 mentioned there are new studies like GTS 2020 and Westerhold et al., 2020 paper that need to be compared and discussed.*

**Reply to Comment #1_ cp-2021-46_ Referee #2:**
See above reply to the same comment in **Comment #2_ cp-2021-46_ Referee #1:**

*#2 Statistical method of testing astronomical signals is also needed.*

**Reply to Comment #2_ cp-2021-46_ Referee #2:**
We added statistical methods for testing the astronomical signals, illustrated in figures 2 and 3 (see detailed ***Reply to Comment #1_ cp-2021-46_ Referee #1***).

*#3 From my understanding, authors anchored the floating time sale to the previously proposed age of EOB and then use this as a starting point to tune the bandpassed 405-kyr of studied data to the orbital solution. This process needs to be clearer in the presentation.*

**Reply to Comment #3_ cp-2021-46_ Referee #2:**
Indeed, this needs clarification because we actually did not tune to the complete orbital solution. We rather tuned to a pure 405 kyr target sine curve, then we anchored the floating time sale to the previously proposed ages of EOB to look for the best phase relationship between the sedimentary NGR and the orbital eccentricity data at the 405 kyr cycle band. Tuning to the complete 400 kyr cycle band from the orbital solution would generate artificial harmonics from the periodic components surrounding the 405 kyr (g2–g5) term.

Subsequently, based on the retained age of EOB that provides the reasonable phase between the NGR and the orbital eccentricity variations (Section 5.1), we adjusted the EOB anchored floating time scale to the orbital solution by tuning only the 405 kyr (g2-g5) cycle extremes in the NGR to their time equivalents in the astronomical signal.

We have now clarified these points in detail in the « Methods » Section 3.3.

*#4 In line 160, "We have aso"should be "we have also"*

**Reply to Comment #4_ cp-2021-46_ Referee #2:**
Corrected.

*#5 In Figure 4, what are the blue lines in panel B and C?*

**Reply to Comment #5_ cp-2021-46_ Referee #2:**
Good catch! This was another (larger) passband for g1-g5 and g2-g5, which provided similar results than the passband depicted by the red curve. We now removed these redundant blue curves from panels B and C.

*If authors can address the above issues, I recommend this as publication.*
*Thank you!*

---

## Author Response (AR2)

Dear Editor,

Please find attached our revised manuscript into two versions: (1) A clean version, and (2) a version with marks showing our revisions (highlighted in pink colored text, and the removed text is crossed out, highlighted in yellow color).

We have considered all revisions you mentioned in the annotated PDF version of the manuscript. We have also revised the 'Abstract' by highlighting the principal results and conclusions of our study. Consequently, the 'Abstract' is now significantly shortened (276 words instead of 395 words in the previous version)

We thank you very much for your availability to handle our manuscript.

Very sincerely,

Slah Boulila

Corresponding co-author